# Did the ERAS Protocol Improve Our Results in Locally Advanced Gastric Cancer Surgery?

**DOI:** 10.3390/jpm12101549

**Published:** 2022-09-21

**Authors:** Özgül Düzgün, Pırıltı Özcan, Mehmet Faik Özçelik

**Affiliations:** 1Department of Surgical Oncology, İstanbul Umraniye Training and Research Hospital, University of Health Sciences, Istanbul 34766, Turkey; 2Department of General Surgery, Cerrahpaşa Faculty of Medicine, İstanbul University, Istanbul 34098, Turkey

**Keywords:** enhanced recovery after surgery, fast-track surgery, gastrectomy, gastric cancer, neoadjuvant, locally advanced

## Abstract

ERAS is a protocol consisting of multidisciplinary approaches, including patient education, fluid resuscitation, keeping patients’ innormothermic state, early feeding, and mobilization. We aimed to present and compare the postoperative results of patients operated on due to locally advanced gastric cancer who received neoadjuvant therapy in two high-volume tertiary centers, and aimed to present that ERAS protocols can be applied with success without increasing the complication rate. This retrospective study consisted of two groups: a non-ERAS group and an ERAS group. The ERAS group consisted of 106 patients and the non-ERAS group consisted of 104 patients. The time to first flatus was shorter in the ERAS group compared with the non-ERAS group: 2.8 (1–5) and 3.5 (1–5, *p* = 0.008), respectively. The average stay at the hospital was 9 (7–22) days in thenon-ERAS group and 6.5 (5–14) days in the ERAS group. Readmission to the hospital within first 15 days after discharge was observed in one (%0.9) patient in the non-ERAS group and in four (%3.8) patients in the ERAS group. ERAS protocols contribute positively to the nutritional status of patients by reducing surgical stress and the rate of hospitalization, and reducing health costs. However, it appears to be associated with increased readmission rates.

## 1. Introduction

Enhanced recovery after surgery (ERAS) protocol was first introduced by Wilmore and Kehlet in 2001, and ever since, it has shown successful positive results in colorectal surgery, and has been successfully practiced in all surgeries involving the gastrointestinal system [1,2]. Enhanced recovery after surgery (ERAS) is a multidisciplinary set of care interventions that optimize perioperative management, and accelerate rehabilitation [3,4]. The superiority of ERAS protocol starts from the preoperative period by the education of patients and by easing the anxiety. The emerging data from both prospective randomized trials and meta analysis studies have supported that effective pain control, early removal of drainage tubes, and early start of oral intake have been more effective during the postoperative period compared to the classical surgical approach [5]. In gastric cancer surgery (either laparoscopic, robotic, or conventional), complete mesogastric excision with D2 lymph node dissection is a must. As the use of minimal invasive surgery has become more prevalent, less pain in the postoperative period and early mobilization have been a great support in the application of ERAS [6,7]. Different versions of ERAS have been practiced in different surgery clinics. However, key elements of ERAS stay the same in every practice. In experienced high-volume centers, early oral intake, early mobilization, and early time to pass flatus have been associated with better outcomes. Along with these, ERAS decrease the length of stay at the hospital and, therefore, the hospital expenses [3,7]. In our study, we aimed to present and compare the postoperative results of patients operated on due to locally advanced gastric cancer (LAGC) with neoadjuvant therapy in two high-volume tertiary centers, and aimed to present that ERAS protocols can be applied with success without increasing the complication rate.

## 2. Materials and Methods

Between June 2017–2022, the data of patients operated on due to LAGC in Istanbul University Cerrahpaşa Medical Faculty General Surgery Clinic and the Health Sciences University Umraniye Research and Training Hospital Surgical Oncology Clinic were collected prospectively and analyzed retrospectively. The study consisted of two groups: a non-ERAS group (Istanbul University Cerrahpasa Medical Faculty General Surgery Clinic) and an ERAS group (Health Sciences University Umraniye Research and Training Hospital Surgical Oncology Clinic). The ERAS group consisted of 106 patients and the non-ERAS group consisted of 104 patients. Total gastrectomy +D2 lymph node dissection was performed on all patients. Written informed consent was obtained from the study participants, and the Ethics Committee of the Health Sciences University Umraniye Research and Training Hospital approved the study (numbered 2022/282). All patients were operated with the approval of the multidisciplinary tumor council. The patients diagnosed with grade 3 gastric cancer who received neoadjuvant therapy were included in this study. In both centers, the operations were performed by the same surgical team. In the non-ERAS group, the patients were not treated with ERAS protocol, and in the ERAS group, the patients were treated with ERAS protocol. Demographic data, comorbidities, Karnofsky performance score, duration of the surgery, amount of bleeding, mobilization time, first oral intake, time to pass first flatus, need for analgesia, complications, discharge time, readmission, and mortality rates were evaluated. Cases without neoadjuvant therapy, emergency cases, and cases that were not grade 3 cancer were excluded from the study.

### 2.1. Statistical Analyses

Statistical analysis was performed using SPSS version 20.0 software (IBMSPSS, Inc., Chicago, IL, USA). The numerical data obtained are summarized in tables as arithmetic mean ± standard deviation, minimum, maximum, and range values. Data were evaluated as frequency and percentages. Continuous variables were expressed as mean ± standard deviation (SD) or median (interquartile range (IQR)), and compared by the independent samples *t*-test or the Mann–Whitney *U*-test. A *p* < 0.05 was regarded as statistically significant.

### 2.2. Preoperative Period

In the ERAS group, patients were informed about the surgery and the expected possible complications and details of the ERAS protocol; this way, we tried to decrease the anxiety level of the patients. Abstaining from alcohol and smoking was advised, and triflow and chewing gum exercises were initiated. Nutritional status was assessed by the nutritional risk score (NRS, 2002). If the total NRS score was higher than 2, a nutrition plan was started. Preoperative nutritional support was initiated with the advice of a nutritionist to patients who had a weight loss of more than 10%. After admission to the surgery clinic, all patients were informed by the ERAS chief nurse. In both groups, low molecular weight heparin (LMWH) was applied to all patient for thromboembolism prophylaxis. In the non-ERAS group, oral intake was stopped 6 h prior to surgery, whereas in the ERAS group, the patients were allowed to drink carbohydrate drinks until 2 h prior to surgery. Bowel preparation was not administered in both groups (Table 1).

### 2.3. Perioperative Period

In both groups, patients were asked to wear pneumatic compression socks in the operation room. The operating room temperature was kept approximately between 18–24 °C with 30–60%humidity. In the ERAS group, a thoracic epidural catheter was used, and long-acting sedative medicine and central venous catheters were avoided. In both groups, antibiotic was applied 30 min prior to the start of surgery. Antibiotics were re-appliedevery 3 h. Heating devices were used in order to preserve normothermia. A loss of volume of liquids, hourly urine output, and arterial pressure were observed during surgery, and necessary fluid resuscitation was applied. In the ERAS group, abdominal drainage tubes were avoided if possible, and in thenon-ERAS group, 2 abdominal drainage tubes were placed.

### 2.4. Postoperative Period

In both groups, the patients were asked to wear anti-embolic socks. LMWH was applied. The use of morphine was avoided in the ERAS group. Pain control was provided by epidural anesthesia. Mobilization was encouraged 4 h after the surgery. Patients were encouraged to walk at least 2 h on the same day of the surgery, and 6 h on the first day post-operation. Urinary catheters were removed on the same day of the surgery. Patients were encouraged to practice triflow and to chew gum. Patients were allowed to drink 1 glass of water6 h after surgery. On post-operative day 1, full liquid fluid intake was allowed if tolerated by the patient. On post-op day 3, a pureed food diet was allowed. Patients were discharged on post-operative day 5 or 6. In the non-ERAS group, a fluid diet was started after the first flatus. Similarly, solid food intake was started after the first defecation. In the ERAS group, abdominal drainage tubes were removed on post-op day 1 or 2, whereas in the non-ERAS group, abdominal drainage tubes were removed on post-op day 6. A phone number was given to the patients in the ERAS group, which they could reach 24/7 in case of emergency or in case they needed consultancy. All patients in both groups were called for polyclinic control 10 days after their discharge from the hospital, and they were routinely called for follow-ups.

## 3. Results

In the non-ERAS group, 106 patients, and in the ERAS group, 104 patients, were operated due to LAGC. Total gastrectomy +D2 lymph node dissection was performed in all patients. In the non-ERAS group, the mean age of the patients were 55 (36–70), 71 (66.9%) of them were male, and 35 (33.1%) of them were female. ASA scores ranged between 1–3, the mean BMI (body mass index) was 22 (19–36), and the mean Karnofsky performance score was 82 (75–100).

In the ERAS group, the mean age of the patients were 57 (33–77), 62 (59.6%) of them were male, and 42 (40.3%) of them were female. ASA scores ranged between 1–3, the mean BMI was 23 (18–34), and the mean Karnofsky performance score was 78 (70–100) (Table 2).

The average operation time was 205.6 (±80.6) and 207.4 (±55.8) minutes in non-ERAS and ERAS groups, respectively. There wasno statistical difference between the two groups regarding the operation time.

In the non-ERAS group, extra organ resection was performed in two patients, whereas in the ERAS group, extra organ resection was performed in five patients. The amount of bleeding was 120.6 (±95.3) and 130.0 (±99.7) mLin the non-ERAS group and ERAS group, respectively. Billroth 2 Roux-n-Y esophagojejunostomy was applied to all patients as reconstruction.

In the ERAS group, patients were encouraged to drink one glass of water on the same day of the surgery, whereas in thenon-ERAS group, oral intake was initiated averagely on post-operative day 3 (3–7). A semi-fluid diet was initiated on post-operative day 2 and 5 in the ERAS group and non-ERAS group, respectively. A soft bland diet was initiated on post-operative day 4 and 6 in the ERAS group and non-ERAS group, respectively. The time to first flatus was shorter in the ERAS group compared with the non-ERAS group: 2.8 (1–5) and 3.5 (1–5, *p* = 0.008), respectively. Except for the time to first flatus, other parameters were not statistically significant (Table 3).

In the non-ERAS group, Clavien–Dindo (CD) grade 3 complication was observed in seven (6.6%) cases, whereas in the ERAS group, CD grade 3 complication was observed eight (7.6%) cases. There was no statistically significant difference regarding complications in both groups. In the non-ERAS group, pleural effusion was observed in two cases, hemorrhage was observed in two cases, anastomosis leakage was observed in two cases, and one case had evisceration. In the ERAS group, anastomosis leakage was observed in two cases, pleural effusion was observed in two cases, intraabdominal abscess was observed in one case, evisceration was observed in one case, and hemorrhage was observed in one case. Interventional radiology placed a catheter in patients with pleural effusion and intraabdominal abscess. A covered stent was placed by gastroenterology to patients with anastomosis leakage. In patients who had evisceration, re-suturing was performed in the operation room. In the ERAS group, one patient was reoperated, and re-esophagojejunostomy was performed. The average stay at the hospital was 9 (7–22) days in the non-ERAS group, and 6.5 (5–14) days in the ERAS group. Readmission to the hospital within first 15 days after discharge was observed in one (0.9%) patient in the non-ERAS group, and in four (3.8%) patients in the ERAS group. These cases were hospitalized due to ileus. Mortality was not observed in the non-ERAS group within 30 days, whereas one patient was considered exitus in the ERAS group due to COVID-19 (Table 4).

## 4. Discussion

ERAS is a protocol consisting of multidisciplinary approaches, including patient education, fluid resuscitation, preventing nausea and ileus, keeping normothermia, early feeding, and mobilization. However, each surgeon has a different experience and educational background. Therefore, many of them may choose to continue with a conventional approach. The complexity of the ERAS protocol, the difficulty in finding a willing and adequate number of staff, the resistance of clinicians for change, and the absence of convincing evidence of the success of ERAS may be the reasons for this. Due to these reasons and difficulties, ERAS practice could find itself a place in tertiary centers with multidisciplinary teamwork. A study from Korea reports that in specific centers specialized in gastric cancer, ERAS is practiced in 30% of patients [8]. In our surgical oncology clinic in Umraniye Training and Research Hospital, we practice ERAS in every case routinely.

Gastric cancer is a surgical procedure in which complications can be observed in up to 45% of patients [4]. Therefore, in clinics where complications are managed with difficulty, ERAS is regarded as taking an unnecessary risk. Patients with many comorbidities, poor oral intake, and who are noncompliant with the program are among the reasons why many surgeons choose to stay distant to this protocol. In fact, all of these reasons are involved in the ERAS basic supporting program. ERAS is practiced commonly in tertiary centers nowadays. These ERAS programs are more commonly practiced in far-Eastern countries, such as China, Korea, and Japan. In these countries, early-stage tumor prevalence is more common, and the patient profile consists of younger patients with less comorbidities, therefore leading to less complication rates. In our country, due to the insufficient screening programs, patients admit to clinics after they are diagnosed with LAGC.

One of the main problems of ERAS is the decision of whether all elements of ERAS are applied or not. Some clinicians may not choose to apply some parameters of ERAS due to their differences in experience. Lombardi et al. reported that in their study consisting of 117 cases, 75% of the cases performed 80% of 24 parameters of ERAS [9]. Rubinkiewicz et al. practiced only 14 of 25 elements of ERAS [10]. FumagalliRomario U. et al. reported in their study that some of the elements of ERAS are not routinely accepted in their practice, especially nutrition screening [4]. Another controversial point is feeding patients with carbohydrates until 2 h prior to surgery. This is not routinely practiced in every clinic. As a matter of fact, Makuuchi et al. reported in their case control study that preoperative carbohydrate loading didnot improve the postoperative nutritional status or maintain the muscle strength post-operatively [11].

In the clinical practice of ERAS, the unnecessary use of analgesics is avoided. In a phase 3 study in which Aoyama et al. compared open vs. minimal invasive surgery, they reported that the extra use of acetaminophen was significantly higher in the open surgery group compared to the minimal invasive surgery group (18.3% vs. 6.2%, p: 0.03) [12].Today, combining minimal invasive surgery with ERAS is associated with lesser pain, an accelerated healing process, and an early return of gastrointestinal functions compared to conventional open surgery [3,4,5,6,7,8,9,10,11,12].Fumagalli et al. practiced minimally invasive surgery to 28% of 270 cases [4].

In clinical practice, the placement of abdominal drains and nasogastric tubes are conventionally used. In the literature, some randomized controlled trials (RCT) and meta-analysis studies show that the use of abdominal drains and nasogastric tubes do not have a supporting or convincing place in gastric surgery [13]. Gianotti et al. defend that there is not enough evidence in order to stop placing abdominal drainage tubes, and not placing abdominal drainage tubes is associated with a higher complication rate [3]. This result contradicts with the other results reported in the literature. In their RCT, Tanaka et al. did not place abdominal drainage tubes in 83.6% of their cases, and reported that abdominal drainage tubes do not decrease the complication prevalence, but increase the risk of intraabdominal collection, infections, and fistulas [14]. Desiderio J et al. placed nasogastric drainage tubes to 35% of their patients, and placed an abdominal drainage tube to 25% of their patients in the ERAS group. In the non-ERAS group, they placed nasogastric drainage tubes and an abdominal drainage tube to all of the patients, and reported similar complication ratios in both groups [15]. In our study, drainage tubes were removed on post-operative day 3 in the ERAS group, and on post-operative day 6 in thenon-ERAS group, and there wasnot a statistically different complication rate between the two groups. According to ERAS protocol, early oral intake can be initiated without any complications. Jeonget al. reported that most patients could resume an oral diet within 1 or 2 days with conservative care, and no serious complication occurred that was associated with early oral nutrition, such as aspiration or anastomosis leakage [13].Jie Ding et al. reported that in meta-analysis studies, a reduction in time to first flatus is an important part of ERAS [16].Rohck et al. achieved their goal of early oral feeding and discharge on the 4th day after surgery in 127 patients with stage 1 and 2a gastric cancer, with a rate of 77% [17]. Li et al. determined that ERAS did not prolong the time to first flatus and did not increase complication rate in their eight RCT and meta-analysis series consisting of 400 cases [18].Liayang et al. reported in their prospective study consisting of 98 cases that oral clear water on post-operative day 1 was accommodated by 86% of patients, oral liquid diet on post-operative day 2 was accommodated by 70% of patients, and oral semi fluid diet on post-operative day 3 was accommodated by 60% of patients [6]. In our study, we found that starting early oral intake in the ERAS group did not increase the risk of leakage, and the time to first flatus was earlier than the non-ERAS group.

It is known that ERAS reduces the hospital stay. Jeong et al. reported that the ERAS protocol resulted in an earlier discharge of 3 days without increasing readmission [8]. In their RCT, Tian Y et al. emphasized that ERAS reduces post-operative pain and complications, causes earlier discharge, and reduces hospital costs [19]. In their RCT, Tanaka R et al. emphasized that theERAS protocol group had a shorter hospital stay, a significantly lower rate of postoperative complications of grade III or higher, and reduced hospitalization cost [14]. In our study, in the ERAS group, the average discharge time was 6.5 days and there was a statistically significant difference between the two groups. We also observed that this corresponds with world literature. One of the biggest concerns in ERAS is the worry that it will increase the complication rates. The benefit of ERAS in reducing postoperative morbidity in gastrectomy is unclear. Previous studies and meta-analyses have shown inconsistent results of ERAS regarding postoperative morbidity after gastrectomy. Jeong et al. found in their study that overall, ERAS did not show a significant difference in morbidity, but showed a trend towards reducing local complications [8]. In the meta-analysis study, Jie Ding reported that ERAS did not increase postoperative complications [16]. Aoyama et al. reported that the ERAS program is safe for both open and minimally invasive surgery without increasing the risk of postoperative complications and mortality [12]. Wang J et al. reported the CD grade 3 complication rate to be 10% [20]. In our study, the CD grade 3 complication rate was 7.6% in the ERAS group and 6.6% in thenon-ERAS group, which corresponds with the literature, and there was no statistically significant difference between the two groups.

Increasing readmission rates is a major concern of ERAS. Jeong et al. and previous studies concerning gastrectomy revealed that the readmission rates in ERAS patients was 0%–6%, and this was not very different compared to conventional care patients [13]. On the other hand, Jie Ding et al. reported that the readmission rates were significantly higher in the ERAS groups than the conventional care group [16]. In the meta-analysis evaluating 18 RCT, Lee Y et al. reported that ERAS significantly increased readmissions (RR 2.43, 95%CI 1.09 to 5.43, p: 0.03) without increasing complication ratios [21]. Wee İJY et al. reported higher readmission rates in the ERAS group as a result of the evaluation of meta-analysis, consisting of 23 articles and 2686 cases [22]. In our study, we found that the readmission rate in the ERAS group (3.8%) was statistically significant compared to the ERAS group. However, these readmissions were due to pain and nutritional problems, and we think that this is due to insufficient education of the patients. Since we did not find a study related to ERAS in LAGC in the literature, we could not present any data to compare the study. All studies in the literature were performed in stage 1–2 gastric cancers. Therefore, we determined that this study on ERAS in LAGC is the first.

The limitations of our study are that firstly, our study was retrospective, the number of cases were low, and the patients were evaluated by different councils in two different institutions. Moreover, in the non-ERAS group, operations were performed by different surgeons (the main operator is the same in all of the operations; however, the residents are different in the non-ERAS group, since it is a university hospital clinic), but in the ERAS group, the operations were performed by the same surgeon (in this group, residents are not involved in operations). In addition, the non-ERAS group was a general surgery clinic, whereas the center performing ERAS was a surgical oncology sub-branch. The non-ERAS group was a university education clinic, and its residents were included in the cases and follow-ups. In the ERAS group, all follow-ups are conducted by subspecialists, and the chemotherapeutics used in the neoadjuvant period are not homogeneous. Lastly, long-term outcomes were not analyzed.

## 5. Conclusions

ERAS protocols in LAGC contribute positively to the nutritional status of patients by reducing surgical stress and the rate of hospitalization, and by reducing health costs. However, it appears to be associated with increased readmission rates. We believe that with the increase of ERAS centers of excellence, better results will be obtained. Higher-quality, high-volumed, multicentered RCTs with long-term follow-ups are needed to more precisely evaluate ERAS protocols in LAGC surgery.

## Figures and Tables

**Table 1 jpm-12-01549-t001:** Perioperative protocols in the ERAS and non-ERASgroups.

Phase	ERAS Protocol Group	Non-ERAS Group
**Preoperative**	Counseling and ERAS education	
	Discuss patient-specific modifications	Discuss patient-specific modifications
	Preassessment for risk adjustment	Preassessment for risk adjustment
	Normal meal at dinner	Normal meal at dinner
	No bowel preparation	No bowel preparation
	Light meal intake until 6 h before surgery Carbohydrate drink until 2 h before surgery	
**Intraoperative**	Intermittent pneumatic compressor	Intermittent pneumatic compressor
	Midthoracic epidural anesthesia and analgesia, infiltration of wound with ropivacaine	Tracheal intubation with general anesthesia
	Combined tracheal intubation and general anesthesia	Avoid hypothermia(<36_C)
	Protective lung ventilation	Routine use of nasogastric tube drainage
	Avoid hypothermia (<36_C)	Routine use of 1–2 abdominal drainage tubes
	No nasogastric tube drainage	
	Target-directed fluid therapy	
	Antibiotic administrated before skin incision, and every 3 h during surgery	Antibiotic administrated before skin incision, and every 3 h during surgery
	Ambulation at evening if possible	
	No routine use of abdominal drainage tube	
	Oral intake of a little clear water after the effects of anesthesia disappear	
**Postoperative**	Continue epidural analgesia for 3 d after surgery	Opioids as main analgesia after surgery
	One antibiotic administration after surgery	Start of ambulation 24 h after surgery
	Training and removal of urine catheter 24 h after surgery	One antibiotic administration after surgery
	Continue ambulation at least four times per day	Intravenous infusion of 2.0–3.0 L of Ringer lactate for 3 d
	Target-directed fluid therapy for 3 d	Mechanical DVT prophylaxis
	PONV prophylaxis (unless contraindicated)	Training and removal of urine catheter 24 h after surgery
	Mechanical DVT prophylaxis	Start to drink water if bowel sounds are heard
	Start of clear liquid diet at dinner on POD1	Diet build-up from the day after flatus; three steps (clear liquid, full liquid, soft diet)
	Start of soft diet asPOD3 is tolerated	

**Abb**: DVT (1–4), deep vein thrombosis; POD (1–4), postoperative day; PONV (1–4), postoperative nausea and vomiting.

**Table 2 jpm-12-01549-t002:** Demographic data of patients.

Characteristics	ERAS Group (*n* = 104)	Non-ERAS Group (*n* = 106)	*p*-Value
Age	57 (33–77)	55 (36–70)	*p* > 0.05
Sex			
Man	62 (59.6%)	71 (66.9%)	*p* > 0.05
Female	42 (40.3%)	35 (33.1%)	*p* > 0.05
BMI	23 (18–34)	22 (19–36)	*p* > 0.05
ASA score	1.9 (1–3)	1.8 (1–3)	*p* > 0.05
Karnofsky Score	78 (70–100)	82 (75–100)	*p* > 0.05
Operation Time (min)	207.4 (±55.8)	205.6 (±80.6)	*p* > 0.05

Data are expressed as mean ± SD and *n* (%). Abb: ASA: American Society of Anesthesiologists; BMI: body mass index.

**Table 3 jpm-12-01549-t003:** Post-operative outcomes.

Characteristics	ERAS Group (*n* = 104)	Non-ERAS Group (*n* = 106)	*p*-Value
First ambulation time	1.1 (1–2)	1.5 (1–3)	*p* >0.05
Removal of foley	1.1 (1–2)	1.2 (1–3)	*p* >0.05
Removal of drains	2 (1–4)	5 (4–7)	*p* >0.05
Sips of water (days)	1.1 (1–2)	3.7 (3–5)	*p* >0.05
Semi-fluid diet (days)	2	5	*p* >0.05
Soft bland diet (days)	4	6	*p* >0.05
Time to first flatus (days)	2.8 (1–5)	3.5 (2–5)	*p*=0.008
Time to first defecation	3.5 (3–5)	5.6 (5–7)	*p* >0.05
Length of stay	6.5 (5–14)	9 (7–22)	*p* >0.05
Readmission	4 (3.8%)	1 (0.9%)	*p* >0.05
Morbidity	8 (7.6%)	7 (6.6%)	*p* >0.05
Mortality	1 (0.9%)	0 (0%)	*p* >0.05

**Table 4 jpm-12-01549-t004:** Post-operative complications by Clavien–Dindograde 3classification.

Complication	ERAS	Non-ERAS	*p*-Value
Pleural effusion	2	2	*p* > 0.05
Hemorrhage	1	2	*p* > 0.05
Anastomotic leak	2	2	*p* > 0.05
Evisceration	1	1	*p* > 0.05
Intraabdominal abscess	2	-	*p* > 0.05
Total	8 (7.6%)	7 (6.6%)	*p* > 0.05

## Data Availability

The datasets generated and analyzed during the current study are available on reasonable request.

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
