# Peer review of "Did the ERAS Protocol Improve Our Results in Locally Advanced Gastric Cancer Surgery?"

_jpm, 2022, doi:10.3390/jpm12101549_

Round 1
Reviewer 1 Report
Summary:
Manuscript JPM-1918586 describes a clinical research study comparing peri-operative outcomes between two patient cohorts undergoing total gastrectomy with D2 lymphadenectomy with and without ERAS protocol in Istanbul, Turkey. The authors showed that length of stay, and return to bowel function were shorter in the ERAS group, however readmission rate was higher. Mortality and short term complications were not different. The authors claim that ERAS contributes to improve overall peri-operative outcomes, decrease surgical stress, and likely cost.
Criticism:
1. Study design: This is a clinical research study comparing two patient cohorts undergoing total gastrectomy with D2 lymphadenectomy for gastric cancer. Both cohorts were operated at different hospitals and it is unclear from an oncological staging standpoint if the two cohorts are comparable. Did the authors register the study with the ERAS(R) society? Discussing nutritional status, do the authors have any biological evidence of the patients' nutritional status? Do the authors have any information of medium term patient outcomes (Event Free and Overall Survival)? Why did the authors only consider complications and readmission within 15 days and not 30 and 90 days as most guidelines and published series usually use these time periods (NSQIP, etc.)?
Introduction and Discussion: The ERAS concept could be better explained in the introduction and its anticipated benefits. The discussion should be edited by a native English speaker and revised thoroughly. There seem to be several Turkish words that remained in the manuscript as well as some significant spelling errors.
Patients and methods: this section is well described and the clear ERAS protocol is well described. Did the authors also use any NSAIDs in their pain management?
Results: The description of results should be clarified. Tables 2 and 3 should include statistical significance in the describe parameters and table 4 should be redrawn so comparison between Eras and Non-Eras cohorts is easier. Complications should also be listed in the same order for both groups.
Conclusion: The study addresses an interesting approach with two large cohorts of patients, however significant questions need to be addressed and the manuscript needs to be edited and reviewed by a native English speaker.
Reviewer 2 Report
Dear Authors,
congratulations for your paper and the data collection. To optimize the quality of your work, I will strongly suggest to correct some structural aspects of your presentation:
1. Could you please describe which criteria were used to select patients for each group?
2. Is there a stratification in those groups describing which surgical techniques (minimally invasive or not) were used and how many of them for each group?
3. Was the surgical technique standardized? If it was, could you please describe it?
Moreover, there are many editing/typing mistakes to be fixed, which certainly will be corrected with more attention.
Best Regards
Round 2
Reviewer 1 Report
The manuscript has been improved however significant issues persist, including grammatical inaccuracies and spelling errors - i.e. Table 3, Duration of first flatsus (days) should be corrected to Time to first flatus (days), and similarly to the other durations to X, Length of stay (days) etc. The p values should be included in details, so the readership can understand if the values were trending towards significance or not and whether power may have been contributing to the absence of statistical significance. In table 4, the authors may want to clarify that none of the complications reached statistical significance.
The discussion has been significantly improved and is very thorough, however as mentioned above several inconsistencies and spelling errors persist.
Author Response
Answers to Reviewer 1:
1.The manuscript has been improved however significant issues persist, including grammatical inaccuracies and spelling errors - i.e. Table 3, Duration of first flatsus (days) should be corrected to Time to first flatus (days), and similarly to the other durations to X, Length of stay (days) etc. The p values should be included in details, so the readership can understand if the values were trending towards significance or not and whether power may have been contributing to the absence of statistical significance. In table 4, the authors may want to clarify that none of the complications reached statistical significance.
Response 1: The article is read again and the grammatical errors were corrected by a native English speaker. Table 3 is corrected as told. In table 4, p values are added in order to clarify that there is not a statsictical siginificant difference among the complications. Also, p values of Table 3 is explained and it is expressed that only time to first flatus was significantly siginificant.
- The discussion has been significantly improved and is very thorough, however as mentioned above several inconsistencies and spelling errors persist.
Response 2: The discussion part is read again and spelling errors are corrected.
Reviewer 2 Report
Dear authors,
thanks for your answer and congratulations for the improvements added to your paper. By the way, I have to stress out another time a crucial point about your statistical population, which was already explicated in my previous comment: how did you select which patient could be enrolled in ERAS group and which not? Because if it was a comparison between two different working groups, with different perspectives about patients' management, the core of your paper results in a different overview.
Furthermore in section "Materials and Methods" you state "In both centers, the operations were performed by the same surgical team", but after, in "Discussion" "The limitations of our study are that firstly, our study was retrospective, the number of cases were low, and the patients were evaluated by different councils in two different institutions. Also, the fact that in non-ERAS group operations were performed by different surgeons, but in the ERAS group the operations were performed by the same surgeon. In addition, Non-ERAS group was a general surgery clinic, while the center performing ERAS was a surgical oncology sub-branch, the non-ERAS group was a university education clinic, and its residents were included in the cases and follow ups".
Please give a feedback about those hints.
Best regards
Author Response
Answers to Reviewer 2:
1.thanks for your answer and congratulations for the improvements added to your paper. By the way, I have to stress out another time a crucial point about your statistical population, which was already explicated in my previous comment: how did you select which patient could be enrolled in ERAS group and which not? Because if it was a comparison between two different working groups, with different perspectives about patients' management, the core of your paper results in a different overview.
Response 1: Thank you for your comment. Umraniye Research and Training Hospital is working as an ERAS center and the patients admitting to our clinic are automatically enrolled to ERAS program. However, in non ERAS group, the other hospital doesn’t apply ERAS protocol therefore patients admitting to that hospital are automatically treated with non ERAS protocol. Therefore, there is no selection among the patients. We only compared the results of two groups in two hospitals.
2.Furthermore in section "Materials and Methods" you state "In both centers, the operations were performed by the same surgical team", but after, in "Discussion" "The limitations of our study are that firstly, our study was retrospective, the number of cases were low, and the patients were evaluated by different councils in two different institutions. Also, the fact that in non-ERAS group operations were performed by different surgeons, but in the ERAS group the operations were performed by the same surgeon. In addition, Non-ERAS group was a general surgery clinic, while the center performing ERAS was a surgical oncology sub-branch, the non-ERAS group was a university education clinic, and its residents were included in the cases and follow ups"
Response 2: Thank you for your comment. In the discussion part, the explanation is made detailly reagrding the operation groups.